# Technical note: An approach for handling multiple temporal frequencies with different input dimensions using a single LSTM cell

Eduardo Acuña Espinoza [1], Frederik Kratzert [2], Daniel Klotz [2, 3], Martin Gauch [4], Manuel Álvarez Chaves [5], Ralf Loritz [1], and Uwe Ehret [1]

[1]Institute of Water and Environment, Karlsruhe Institute of Technology (KIT), Karlsruhe, Germany
[2]Google Research, Vienna, Austria
[3]Machine Learning in Earth Science, Interdisciplinary Transformation University Austria, Linz, Austria
[4]Google Research, Zurich, Switzerland
[5]Stuttgart Center for Simulation Science, Statistical Model-Data Integration, University of Stuttgart, Stuttgart, Germany

**Correspondence:** Eduardo Acuña Espinoza (eduardo.espinoza@kit.edu)

**Abstract.** Long Short-Term Memory (LSTM) networks have demonstrated state-of-the-art performance for rainfall-runoff hydrological modeling. However, most studies focus on daily-scale predictions, limiting the benefits of sub-daily (e.g. hourly) predictions in applications like flood forecasting. Moreover, training an LSTM exclusively on sub-daily data is computationally expensive, and may lead to model-learning difficulties due to the extended sequence lengths. In this study, we introduce a new architecture, multi-frequency LSTM (MF-LSTM), designed to use input of various temporal frequencies to produce sub-daily (e.g. hourly) predictions at a moderate computational cost. Building on two existing methods previously proposed by coauthors of this study, the MF-LSTM processes older inputs at coarser temporal resolutions than more recent ones. The MF-LSTM gives the possibility to handle different temporal frequencies, with different number of input dimensions, in a single LSTM cell, enhancing generality and simplicity of use. Our experiments, conducted on 516 basins from the CAMELS-US dataset, demonstrate that MF-LSTM retains state-of-the-art performance while offering a simpler design. Moreover, the MF-LSTM architecture reported a 5x reduction in processing time, compared to models trained exclusively on hourly data.

## 1 Introduction

Data-driven methods, particularly Long Short-Term Memory networks (LSTMs) (Hochreiter and Schmidhuber, 1997), have demonstrated state-of-the-art performance in rainfall-runoff hydrological modeling (Kratzert et al., 2019b; Lees et al., 2021; Loritz et al., 2024). Currently, most studies primarily focus on daily-scale predictions. However, certain applications, such as flood forecasting, can benefit from sub-daily scale predictions, especially in small fast-responding catchments. These higher temporal resolutions allow the model to better capture an event's magnitude and avoid artificial attenuation or dampening caused by daily aggregation. In addition, they allow the model to reproduce more accurately the temporal dynamics of the hydrograph, and open up the possibility to capture flash floods. For this reason, many operational flood forecasting services, including the National Water Prediction Service of the National Oceanic and Atmospheric Administration (NOAA) in the US

and the Flood Forecasting Center Baden-Württemberg (HVZ) in Germany, produce forecasts at a sub-daily resolution for their operational services.

One major drawback of running LSTM models at exclusively hourly resolution is the significant increase in computational cost for both model training and evaluation. For instance, studies using daily-resolution LSTM models typically employ a sequence length of 365 days for predictions (Kratzert et al., 2019b; Klotz et al., 2022; Lees et al., 2021; Loritz et al., 2024). By spanning a full year of data, this approach allows the LSTM model to capture long-term seasonal processes, such as snowmelt (Kratzert et al., 2019a). However, for hourly data, the equivalent sequence length increases to $365 \times 24 = 8\,760$ timesteps, leading to a substantial increase in the computational resources required. Moreover, LSTMs have shown difficulties in learning information over long sequence lengths (Chien et al., 2021; Zhang and You, 2020), which would create a direct limitation when working exclusively with high-frequency data, such as hourly or 15-min resolutions.

A potential strategy to tackle this problem is to reduce the sequence length. However, this comes at the cost of excluding long-term processes. For example, if a sequence length of 365 timesteps is maintained when working with hourly data, the look-back period would only cover two weeks, as opposed to a full year. Consequently, the model might not account for important long-term dynamics.

Another possible solution is the concept of ODE-LSTMs, proposed by Lechner and Hasani (2020). The authors handle non-uniformly sampled data by the use of a continuous-time state representation of recurrent neural networks. Gauch et al. (2021) carried out experiments exploring the potential of ODE-LSTM for rainfall-runoff modelling, however, they indicate that this method achieved lower performance at a higher computational time than their proposed alternative.

Gauch et al. (2021) proposed the idea of processing older inputs at coarser temporal resolutions compared to more recent data. This approach is based on the fact that, for a dissipative system like a catchment, the importance of the temporal distribution of inputs diminishes the further back in time we look (Loritz et al., 2021). For instance, in cases where discharge during spring is driven by snow melt, the exact hour in which snow accumulated two months earlier is unlikely to affect the hydrograph. Similarly, when modelling a storm, the basin's response will vary depending on soil saturation. If the soil is saturated due to heavy rain over the past month, the precise timing of a peak in rainfall three weeks ago becomes irrelevant. Thus, this approach of handling inputs at different temporal resolutions allows the model to capture long-term processes without the computational burden of processing all data at high frequency. In the following, we use a concrete example to both better illustrate the ideas proposed by Gauch et al. (2021) and to make the connection with our method. For this, we will use one year of data to make a prediction, but only the most recent two weeks ($14 \times 24 = 336$ timesteps) will be processed at hourly resolution, while the rest is processed at daily resolution. The number of timesteps processed at each frequency is a model hyperparameter.

The first architecture proposed by Gauch et al. (2021), referred to as sMTS-LSTM, begins with a forward pass at daily resolution (e.g., 365 timesteps). The LSTM's hidden and cell states from two weeks prior to the final timestep are then retrieved, and the LSTM is reinitialized with these states. Then a second forward pass is performed, using hourly data for the last two weeks. Moreover, since both daily (from the first forward pass) and hourly (from the second forward pass) predictions are

available for the last two weeks, the authors proposed a regularization technique, in which an extra term is incorporated in the loss function to induce consistency between the daily and hourly predictions.

One limitation of this architecture, highlighted by the authors, is that because the same LSTM cell processes both daily and hourly data, the input at both timescales must include the same number of variables. As they mentioned, this can be problematic in operational settings where different temporal resolutions often have different available variables. To address this, the authors proposed a more general architecture called MTS-LSTM. In this variant, the hidden and cell states retrieved from two weeks prior are passed through a transfer function and the result is used to initialize a second LSTM cell, which processes the hourly data. The advantage of this approach is that, with separate LSTM cells for each temporal frequency, different sets of input variables can be used at each resolution. We refer to Fig. B1 for a graphic visualization of these ideas.

Building on the work of Gauch et al. (2021), we propose a new methodology that combines the strengths of both models. We refer to it as multi-frequency LSTM (MF-LSTM). On one hand, this new methodology retains the simplicity and elegance of the sMTS-LSTM by using a single LSTM cell to process data at multiple temporal frequencies. On the other hand, we keep the ability of the MTS-LSTM to handle different numbers of input variables at each frequency, which we accomplish through the use of embedding layers. Moreover, and as explained in detail in the following sections, we make predictions only on the highest frequency (e.g. hourly), and the remaining frequencies are recovered by aggregation, which guarantees cross-timescale consistency without the use of additional regularization.

The remainder of the manuscript is structured as follows. Section 2 details the MF-LSTM architecture and the experimental setup, including the datasets used and the benchmark comparisons. In Section 3, we present and analyze the results of these experiments. Finally, Section 4 summarizes the key findings and offers the study's conclusions.

## 2    Data and methods

### 2.1    Data and benchmarking

To ensure consistency with Gauch et al. (2021) and to enable a direct comparison, we followed their experimental setup. Keeping the same experimental setup allowed us to compare their results against our proposed method, without having to rerun their experiments. The importance of driving model improvement through community benchmarks has been previously discussed in the machine learning and the hydrological community (Donoho, 2017; Nearing et al., 2021; Klotz et al., 2022; Kratzert et al., 2024).

The experiments were conducted on 516 basins located across the contiguous United States, all of which are part of the CAMELS-US dataset (Addor et al., 2017). From this dataset, we extracted 26 static attributes (see Table A3). The hourly input data (see Table A1) were extracted from NLDAS-2 hourly products (Xia et al., 2012), while the target discharge data was retrieved from the USGS Water Information system (USGS, 2016). Following standard machine learning practices, the data was divided into three subsets. The training period spanned from 1990/10/01 to 2003/09/30, the validation period from 2003/10/01 to 2008/09/30, and the testing period from 2008/10/01 to 2018/09/30.

## 2.2 MF-LSTM: multi-frequency LSTM

The concept of MF-LSTM comes from the principle that an LSTM cell has no inherent limitation for processing data at different temporal frequencies. In contrast to process-based hydrological models, where one would not update a storage (say interflow) with a 5mm/h flux (say evapotranspiration) in one timestep and then with 8mm/day in another, an LSTM can accommodate an equivalent updating scheme. For processing an input an LSTM always processes one step in a sequence. However, there is no explicit assumption about the progress of time within one such step. Due to its time-dependent gating mechanisms, the LSTM can learn to modulate how the cell states are updated, regardless of the temporal resolution of the inputs. Consequently, we can leverage this property to handle multiple temporal frequencies within a single LSTM cell, processing older inputs at coarser resolutions and more recent data at higher resolutions.

A concrete example of this approach is illustrated in Fig. 1a, using daily and hourly frequencies. In this example, our goal is to predict hourly discharge. To capture long-term processes, we initially input a full year of high-resolution hourly data (e.g. $365 * 24 = 8760$ hourly timesteps). To avoid the computational burden and learning difficulties associated with processing long sequences, the first $n$ time steps were processed at a coarser resolution, reducing the length of the input sequence entering the LSTM cell. The number of timesteps processed at each resolution is a model hyperparameter and can be determined through hyperparameter tuning.

The example in Fig. 1a shows the case where the first $n = 351 * 24 = 8424$ timesteps, were aggregated into 351 blocks, each containing the average of 24 hourly measurements. Given the normalization of the input and target data and the non-mass-conservative structure of the LSTM, both the average or the sum of the hourly measurements can be used. The remaining $m = 14 * 24 = 336$ timesteps were then processed at the original hourly resolution. By applying this method, we reduced the original sequence length from 8760 to $351 + 336 = 687$ timesteps, decreasing the amount of data to be processed by a factor of 12.8.

To inform the LSTM about the frequency it should be operating on we added a flag channel. This has a value of zero for the first 351 timesteps and a value of one for the remaining 336. Adding a flag to help the model distinguish between different types of conditions is a common practice in machine learning, as it provides the model with additional context (Nearing et al., 2022). For the LSTM specifically, the flag channel acts as an additional bias, that further regulates the gating mechanisms. Fig 1b shows the inclusion of the flag channel for the different frequencies.

Note that in the previous paragraph, we used predefined values to simplify the explanation and clarify the concept. However, the method is by no means restricted to this setup, and its flexibility allows it to adjust the number of time steps processed at each resolution and the order in which the different frequencies are applied. Additionally, the composition of the time series can also be alternated from batch to batch during training or inference. Moreover, the method does not have a restriction to use only two frequencies, and as we show in the next section, a weekly-daily-hourly frequency scheme can be handled without any additional burden.

One of the main advantages stated by Gauch et al. (2021) about the MTS-LSTM architecture is its ability to handle a variable number of inputs for each frequency, because different LSTM cells are used for each temporal frequency (see Fig.

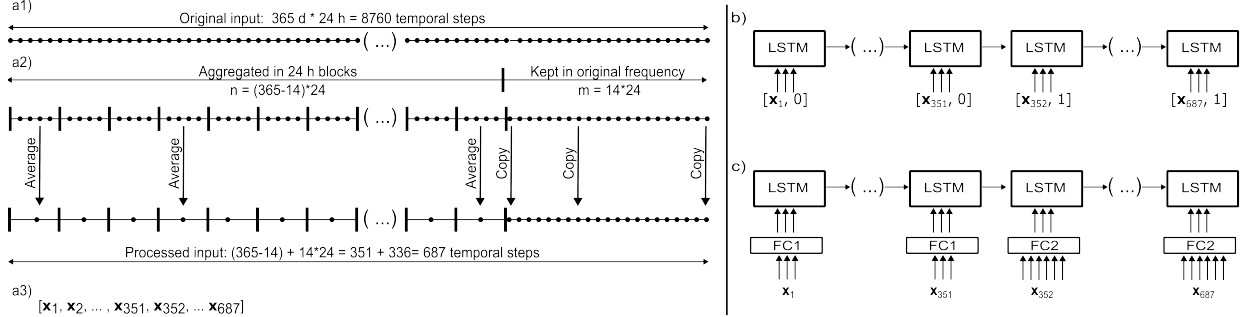

**Figure 1.** Data handling structure for MF-LSTM. a1) The original sequence length consists of one year of hourly data: 8760 temporal steps. a2) The first $(365-14)*24 = 351*24 = 8424$ timesteps are aggregated into 351 blocks, while the remaining $14*24=336$ timesteps are processed in their original hourly frequency. a3) The final input series that will be processed by the LSTM cell consists of $351+336=687$ timesteps. b) In the case where the same number of inputs for each frequency is available, we add a flag as an additional channel to help the LSTM identify each frequency c) In the case where a different number of inputs for each frequency is available, a fully connected linear layer (FC) can be used to map the variable number of inputs of each frequency to a predefined number of channels.

B1). We propose the use of embedding networks as an alternative solution. By using one embedding network for each temporal frequency, we can map different numbers of inputs into a shared dimension. This strategy allows us to separate the steps of our pipeline. We use the LSTM only for sequence processing, and we use the embedding networks to prepare the original information into the format/type that the LSTM cell requires. In the simplest case, the embedding networks could even be a single linear layer, as will be used for the rest of this paper (see Fig. 1c). We evaluated the embedding network with and without the flag channel and observed comparable performance in both cases. This result indicates that the embedding network can internally identify frequency information without the need for an additional flag channel. Therefore, we opted for the simpler approach and excluded the extra channel.

Summarizing, the main distinction between MF-LSTM and sMTS-LSTM lies in MF-LSTM's ability to handle a different number of inputs for each temporal frequency, which gives an advantage in operational settings where different temporal resolutions often have different available variables. Moreover, the primary difference between MF-LSTM and MTS-LSTM lies in the simpler architecture of the former, which uses a single LSTM cell in contrast to one cell per frequency. This results in a more parsimonious model that aligns closely in structure and usage with traditional single-frequency LSTM models.

## 3 Results and Discussion

### 3.1 Performance comparison

Our long-term goal, which goes beyond the scope of this study, is to implement an operational hourly hydrological forecasting system using machine learning methods. The MF-LSTM method is a step toward achieving this, as it enables the computationally efficient simulation of hourly discharges, while allowing to handle a variable number of inputs at each temporal

resolution—both requirements for our broader objective. Consequently, the results reported in this section will focus on two aspects: the ability of the MF-LSTM to produce hourly discharge, and the ability to handle a variable number of inputs while doing so.

Gauch et al. (2021) presented two experimental setups that address these aspects. Therefore, we ran these experiments as a benchmark, to evaluate the performance of our method against their results. Both experiments evaluate the case in which one is interested in simulating hourly discharges, and do so by processing part of the information in daily frequency and part of it in hourly. More specifically, one year of data is processed, but only the last 14 days (336 hours) are processed in an hourly frequency. The value of 336 hours was identified in the original study by hyperparameter tuning. In all cases, the results are reported using an ensemble of 10 independent LSTM models, that were initialized using different random seeds. The final simulation value is taken as the median streamflow across the 10 models for each timestep.

The first experiment evaluated the scenario where the same number of inputs (see Table A1) were used for both daily and hourly processing. In this case, we can directly compare our model's performance with the results reported by Gauch et al. (2021) for the MTS-LSTM, sMTS-LSTM, and also what they refer to as the Naive approach. The Naive approach involves running a standard LSTM exclusively on hourly data, with a sequence length of 4 320 hours (6 months). We can see from Fig. 2a that all the models present the same performance up to the second decimal, with a median NSE of 0.75.

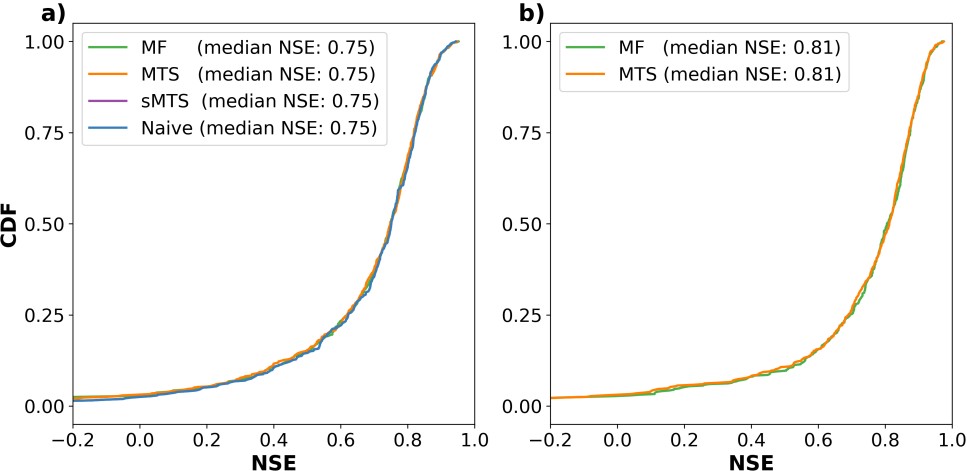

**Figure 2.** Cumulative NSE distribution of the different models, evaluating the prediction accuracy for hourly discharges along 516 basins in the US. a) Case where the same number of variables is used during daily and hourly processing (see Table A1), b) Case where different numbers of variables are used during daily and hourly processing (see Table A2).

The second experiment evaluated the scenario in which different numbers of inputs were used for the daily and hourly steps (see Table A2). Specifically, the daily frequency incorporated 10 dynamic variables from the Daymet and Maurer forcing datasets, while the hourly steps included 21 variables. Eleven of these variables were sourced from the NLDAS forcing at hourly resolution, and the remaining 10 were a low-resolution rediscretization of the 10 daily variables into an hourly frequency (i.e.,

the daily value was repeated 24 times). Consistently with Gauch et al. (2021), since the Maurer forcings range until 2008, the results of this experiment are reported for the validation period. As shown in Figure 2b, the MF-LSTM achieves performance comparable to that of the MTS-LSTM, both reporting a median NSE of 0.81. Comparisons with the sMTS and Naive models are not possible for this experiment, as previously explained, because these models cannot accommodate different numbers of variables for different frequencies.

The previous experiments showed that the MF-LSTM architecture can achieve state-of-the-art performance, fully comparable with the MTS-LSTM and sMTS-LSTM architectures. The results show that a single LSTM cell can handle multiple temporal frequencies at the same time. Moreover, the second experiment corroborates that a simple fully connected linear layer can successfully encode different numbers of input variables into a predefined number of channels.

We also ran an additional experiment to evaluate the capacity of the model to handle more than two frequencies. Specifically, we used a weekly-daily-hourly scheme. The first half of the year (182 days) was handled using a weekly aggregation. The next five and a half months (169 days) at daily resolution, and the remaining 14 days used an hourly frequency. Our results showed that the MF-LSTM is capable of handling this case, presenting a similar performance (see Fig. C1), and reducing the sequence length from 687 to 531.

## 3.2 Computational efficiency

As shown in Fig.1a, one key advantage of processing older inputs at coarser temporal resolution than more recent ones is the reduced computational cost, particularly when compared with feeding in the whole sequence length at a finer resolution (e.g. hourly). This reduction in computational cost impacts not only training time but also memory usage. With long sequence lengths, one might run out of GPU memory, or be forced to take alternative strategies such as reducing the batch size during training and evaluation.

However, the total training time is influenced by external factors, such as differences in hidden size or batch size, which are not directly related to the methods themselves. To minimize these external effects, we conducted an additional experiment where we standardized the hidden size and batch size across all models and compared the average time needed to process a batch. The results showed that MF-LSTM, MTS-LSTM, and sMTS-LSTM exhibited nearly identical efficiency, while the Naive approach was approximately five times slower. For reference, training the MF-LSTM on a Tesla V100 GPU took around seven hours.

## 4 Conclusions

In this study, we introduced the MF-LSTM architecture, designed to produce sub-daily (e.g. hourly) predictions at a moderate computational cost, while giving the model access to long sequences of input data. Building on Gauch et al. (2021) our method processes the input's temporal sequence using different aggregations. Hence, it accounts for the fact that the effect of the input's temporal distribution diminishes the further we look back in time. This allows the MF-LSTM to predict hourly discharges without the overhead of handling the entire sequence at a fine temporal scale.

The ability of the LSTM to maintain performance while handling data from the past at lower resolutions highlights how the LSTM cell is acting similarly to a process-based hydrological model, with a dissipative behavior when it comes to memory of past forcings. This is a step towards understanding LSTM based predictions better they are gaining popularity for applications in hydrology.

As high-resolution data becomes increasingly available in environmental sciences, traditional LSTM models will continue to face challenges when trying to learn from these long sequence lengths. The approach we present here, with its simplicity and computational efficiency, offers a practical solution. Areas like weather forecasting, where minute-resolution data is not uncommon, might benefit from this type of model. Moreover, the possibility to combine multiple frequencies, like our weekly-daily-hourly scheme, enables modelers to extend look-back periods. This may also be beneficial in other domains such as groundwater, where long-term historical data is required to capture slow dynamic processes.

Our proposed embedding strategy opens the possibility to map different numbers of inputs into a shared dimension. This flexibility not only simplifies the model architecture by allowing a single LSTM cell to handle diverse input configurations but also enhances the model's adaptability in operational settings, where the availability of input data may vary across time scales. This overcomes the limitation previously stated in the sMTS-LSTM.

Furthermore, we demonstrate that a single LSTM cell can effectively manage processes operating at different time scales, eliminating the need for separate LSTM cells for each time scale, plus transfer functions between their hidden states. This results in a more parsimonious model that aligns more closely in structure and usage with traditional single-frequency LSTM models, making the transition from single-frequency to multi-frequency LSTMs more intuitive for users.

Through experiments on 516 basins from the CAMELS-US dataset, the MF-LSTM demonstrated the same performance to the MTS-LSTM and sMTS-LSTM models, indicating that the added simplicity and generality does not come at the expense of predictive capability. Moreover, the new architecture presents a similar computation cost as the two previous options and reduces the training time by a factor of 5 when compared to the Naive approach.

The fact that a single LSTM cell allows us to handle multiple frequencies could be attributed to the high similarities between processes at different timescales (e.g., daily and hourly). The LSTM architecture takes advantage of these similarities, along with its ability to regulate gates based on the current context, enabling it to effectively process multiple frequencies. By using a single LSTM cell, we can leverage the additional information content encoded in our data.

The hyperparameters of the model were adopted from Gauch et al. (2021), who conducted hyperparameter tuning. We acknowledge that transferring these parameters across different architectures may not guarantee optimal model performance. However, the primary objective of this technical note is to introduce the new architecture. Furthermore, we demonstrate that, even with the given hyperparameters, the proposed model achieves performance comparable to the current state-of-the-art.

## Appendix A:  Additional information of experimental design

220

The following tables present the variables used in the experiments associated with this study. Table A1 and Table A2 present the variables used in the first and second experiments respectively. The third and fourth columns of each table indicate if the variable was used in the daily frequency, hourly frequency or both. Table A3 names the 26 static attributes used as additional input in models.

**Table A1.** Dynamic input variables used in first experiment, where the same number of variables is used for the daily and hourly frequency.

| Variable Name | Forcing | Daily frequency | Hourly frequency |
|---|---|:---:|:---:|
| convective_fraction | NLDAS Hourly | ✓ | ✓ |
| longwave_radiation | NLDAS Hourly | ✓ | ✓ |
| potential_energy | NLDAS Hourly | ✓ | ✓ |
| potential_evaporation | NLDAS Hourly | ✓ | ✓ |
| pressure | NLDAS Hourly | ✓ | ✓ |
| shortwave_radiation | NLDAS Hourly | ✓ | ✓ |
| specific_humidity | NLDAS Hourly | ✓ | ✓ |
| temperature | NLDAS Hourly | ✓ | ✓ |
| total_precipitation | NLDAS Hourly | ✓ | ✓ |
| wind_u | NLDAS Hourly | ✓ | ✓ |
| wind_v | NLDAS Hourly | ✓ | ✓ |

**Table A2.** Dynamic input variables used in second experiment, where different number of variables is used for the daily and hourly frequency.

| Variable Name | Forcing | Daily frequency | Hourly frequency | Note |
|---|---|:---:|:---:|---|
| prcp(mm/day) | Daymet daily | ✓ | ✓ | *LRR |
| srad(W/m2) | Daymet daily | ✓ | ✓ | *LRR |
| tmax(C) | Daymet daily | ✓ | ✓ | *LRR |
| tmin(C) | Daymet daily | ✓ | ✓ | *LRR |
| vp(Pa) | Daymet daily | ✓ | ✓ | *LRR |
| prcp(mm/day | Maurer daily | ✓ | ✓ | *LRR |
| srad(W/m2) | Maurer daily | ✓ | ✓ | *LRR |
| tmax(C | Maurer daily | ✓ | ✓ | *LRR |
| tmin(C) | Maurer daily | ✓ | ✓ | *LRR |
| vp(Pa) | Maurer daily | ✓ | ✓ | *LRR |
| convective_fraction | NLDAS Hourly | - | ✓ | |
| longwave_radiation | NLDAS Hourly | - | ✓ | |
| potential_energy | NLDAS Hourly | - | ✓ | |
| potential_evaporation | NLDAS Hourly | - | ✓ | |
| pressure | NLDAS Hourly | - | ✓ | |
| shortwave_radiation | NLDAS Hourly | - | ✓ | |
| specific_humidity | NLDAS Hourly | - | ✓ | |
| temperature | NLDAS Hourly | - | ✓ | |
| total_precipitation | NLDAS Hourly | - | ✓ | |
| wind_u | NLDAS Hourly | - | ✓ | |
| wind_v | NLDAS Hourly | - | ✓ | |

*LRR: low-resolution rediscretization is done when the original daily value is used in hourly frequency. Therefore, the original daily value is repeated 24 times.

**Table A3.** Name of the 26 static attributes used in the experiments

| | | | |
|---|---|---|---|
| elev_mean | slope_mean | area_gages2 | frac_forest |
| lai_max | lai_diff | gvf_max | gvf_diff |
| soil_depth_pelletier | soil_depth_statsgo | soil_porosity | soil_conductivity |
| max_water_content | sand_frac | silt_frac | clay_frac |
| carbonate_rocks_frac | geol_permeability | p_mean | pet_mean |
| aridity | frac_snow | high_prec_freq | high_prec_dur |
| low_prec_freq | low_prec_dur | | |

 **Appendix B:  Structure of MTS-LSTM model architecture**

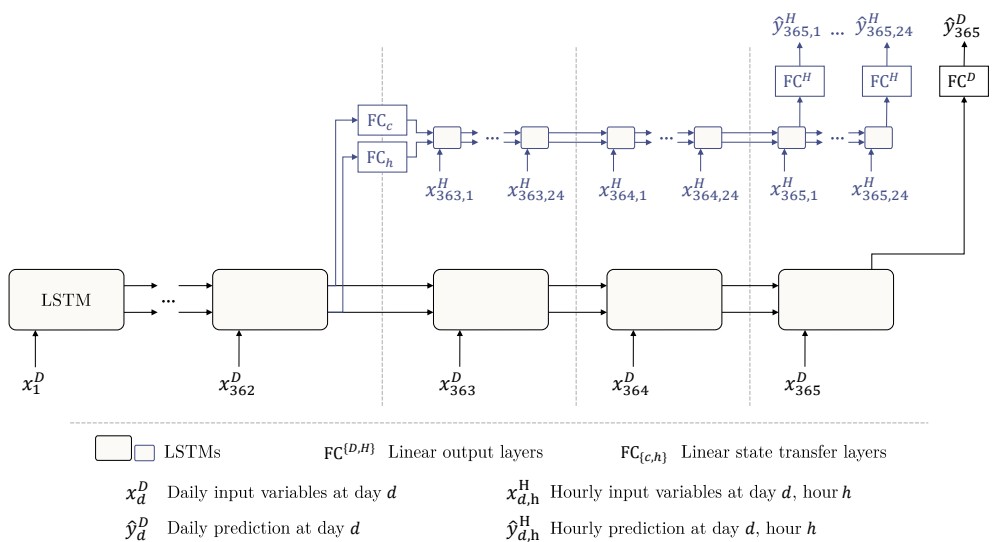

**Figure B1.** Illustration of the MTS-LSTM architecture that uses one distinct LSTM per timescale. In the depicted example, the daily and hourly input sequence lengths are $T^D = 365$ and $T^H = 72$ (we chose this value for the sake of a tidy illustration; the benchmarked model uses $T^H = 336$). In the sMTS-LSTM model (i.e., without distinct LSTM branches), $FC_C$ and $FC_h$ are identity functions, and the two branches (including the fully connected output layers $FC^H$ and $FC^D$) share their model weights.

**Source:** This figure and its description were taken from Gauch et al. (2021).

## Appendix C: Additional results

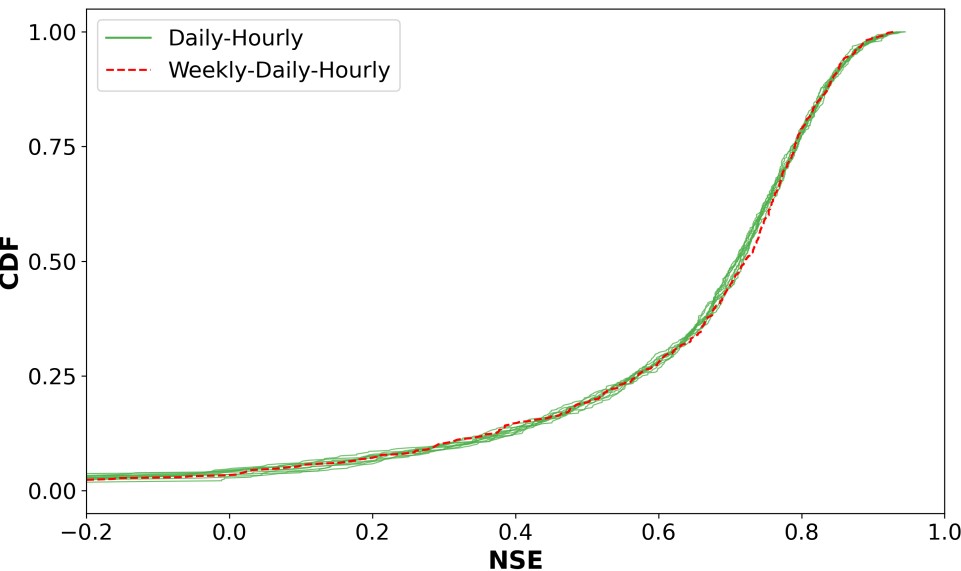

**Figure C1.** Comparison of cumulative NSE distributions for different frequency sequences. The daily-hourly experiment includes 10 distributions, each corresponding to an ensemble member generated through different random initializations. The average of the 10 median NSE values is 0.71. In contrast, the weekly-daily-hourly experiment consists of a single simulation, yielding a median NSE of 0.72.

*Code availability.* The code used for all analyses in this paper is publicly available at https://doi.org/10.5281/zenodo.14780059 (Acuna Espinoza, 2025). It is part of the Hy2DL library, which can be accessed on GitHub: https://github.com/eduardoAcunaEspinoza/Hy2DL.

*Data availability.* All the data generated for this publication can be found at https://doi.org/10.5281/zenodo.14780059 (Acuna Espinoza,
230   2025). The benchmark models can be found at https://doi.org/10.5281/zenodo.4095485 (Gauch et al., 2020b). The hourly NLDAS forcing and the hourly streamflow can be found at https://doi.org/10.5281/zenodo.4072701 (Gauch et al., 2020a). The CAMELS US dataset can be found at https://doi.org/10.5065/D6G73C3Q (Newman et al., 2022). However, one should replace the original Maurer forcings with the extended version presented in https://doi.org/10.4211/hs.17c896843cf940339c3c3496d0c1c077 (Kratzert, 2019)

*Author contributions.* The original idea of the manuscript was developed by F.K., M.G., and D.K. The codes were written by E.A.E. The
235   simulations were conducted by E.A.E. Results were further discussed by all authors. The draft of the manuscript was prepared by E.A.E. Reviewing and editing was provided by all authors. Funding was acquired by U.E. All authors have read and agreed to the current version of the manuscript.

*Competing interests.* Some authors are members of the editorial board of HESS.

*Acknowledgements.* We would like to thank the Google Cloud Program (GCP) team, for awarding us credits to support our research and run
240   the models. The authors also acknowledge support by the state of Baden-Württemberg through bwHPC. UE would like to thank the people of Baden-Württemberg who, through their taxes, provide the basis for this research.

*Financial support.* This work was supported by the KIT Center MathSEE and the KIT Graduate School for Computational and Data Science under the EPO4Hydro Bridge PhD grant.

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
