# Peer review of "Technical note: An approach for handling multiple temporal frequencies with different input dimensions using a single LSTM cell"

_EGUsphere, 2024_

## Author Comment (AC1)

**Response to RC1: 'Comment on egusphere-2024-3355', Anonymous Referee #1**

We want to thank the referee for the detailed evaluation of our paper. In this document, we answer the questions, comments and suggestions given. We will address those comments individually. For clarity, the original comments posted by the referee are written in blue.

The paper addresses the challenge of predicting sub-daily forecasts. In such cases, sub-daily inputs are utilized to achieve optimal performance. However, when longer dependencies are present, processing this data at a sub-daily resolution can be quite time-consuming, as both sub-daily and monthly information may be required.

The authors introduce a simple and innovative approach to handle both short and long dependencies using the same LSTM model. They demonstrate that LSTM can effectively manage data with different frequencies by incorporating a label that indicates the data frequency, without sacrificing performance. Additionally, they show that LSTM can accommodate varying numbers of inputs at different frequencies by including an embedding layer before the LSTM.

These findings apply to any forecasting problem involving multiple time dependencies, suggesting that the proposed approach could have widespread utility.

The paper is well-written, with clear results, and I believe it should be accepted with minor comments.

We thank the referee for the well-structured summary of our paper.

Minor comments:

Line 25-26: I believe that one year is insufficient to capture groundwater behavior due to the longer residence times in these systems. Even in snowmelt-dominated catchments, additional memory may be necessary if snow accumulates between years. If you wish to retain this sentence, you must include a reference to support this assertion or refrain from mentioning specific processes.

Response: Thank you for the suggestion. We agree that groundwater processes might have residence times longer than a year. For snow-dominated catchments, Kratzert et al. (2019) show an example of a snow influence basin, where they demonstrated that the output of the LSTM during the snowmelt period considers input up to 150 days prior. Therefore even though in some basins multiannual snow accumulation might play a role, in most cases one year of data might be enough. We propose to modify the sentence as such:

> *By spanning a full year of data, this approach allows the LSTM model to capture long-term seasonal processes, such as snowmelt (Kratzert et al., 2019).*

Line 98: It would be helpful to provide a brief explanation of the example before presenting any values. For example, Why are you using 351?

Response: In line 98, our idea is to explain the architecture with a concrete example, to make it easier to understand. We used the same values as the ones used during the actual experiments to increase consistency, however, the values in the example could be arbitrary. The value of 351 timesteps at daily resolution and 336 at hourly resolution were taken from Gauch et al. (2021), where they used hyperparameter tuning to determine them. We will add this explanation to a revised version of the manuscript.

Line 106-107: This section indicates that the value of 351 is arbitrary and that any other value could be used. If this is the case, does it imply that this value is a hyperparameter? How should it be estimated? Additionally, how do you determine the duration when dealing with hourly, daily, and monthly periods?

Response: Yes exactly, this is a hyperparameter. Given the computation cost of training the models, the hyperparameter tuning step can be highly time-consuming. Gauch et al. (2021) applied a grid search method, which could also be applied here. Another option is to use surrogate-assisted optimization, however, this method is more complex, and one has to build a high-fidelity surrogate model, which can become a topic of its own.

For the hourly-daily-weekly experiment, we maintained the one-year sequence length and we also processed the last 14 days in hourly resolution. The cut between the amount of data that was processed weekly and daily was done somehow arbitrarily.

Line 159: You mentioned that the median KGE was similar, but what about the entire distribution (CDF)? If there are no significant differences, you could include the figure in the appendix. Did you consider extending the sequence beyond one year, particularly since you can now process longer sequences with reduced computational costs?

Response: Thank you for the suggestion. We will add a figure with the full distribution in a revised version of the manuscript.

About including sequence length longer than a year, this would depend on the type of application one is interested in. However, it is important to clarify that if one is interested in certain processes that might require longer lookback periods, the experiments should be designed accordingly. In our case, we are using a variation of the mean squared error loss function to train our model, which implicitly prioritizes high flows. These high flows are usually explained well enough with a lookup period of one year. If one is interested in groundwater processes that might require multi-annual lookback periods, the loss function and selection of the training points should reflect this. Otherwise, even with longer lookback periods, the model would not be able to utilize the additional information.

Therefore, we agree that the MF-LSTM architecture opens up the possibility of extending the sequence lengths without incurring on a prohibited computational cost. However, depending on the objective of the experiment, other factors should also be considered.

Final remarks

We would like to thank the referee for the overall positive evaluation of our manuscript and hope we could address the questions raised in a satisfactory manner.

References

- Gauch, M., Kratzert, F., Klotz, D., Nearing, G., Lin, J., and Hochreiter, S.: Rainfall–runoff prediction at multiple timescales with a single Long Short-Term Memory network, Hydrology and Earth System Sciences, 25, 2045–2062, https://doi.org/10.5194/hess-25-2045-2021, 2021.

- Kratzert, F., Herrnegger, M., Klotz, D., Hochreiter, S., & Klambauer, G. (2019). NeuralHydrology -- Interpreting LSTMs in Hydrology. *arXiv.* https://doi.org/10.48550/arXiv.1903.07903

---

## Author Comment (AC2)

**Response to RC2: Comment on egusphere-2024-3355, Anonymous Referee #2**

We want to thank the referee for the detailed evaluation of our paper. In this document, we answer the questions, comments and suggestions given. We will address those comments individually. For clarity, the original comments posted by the referee are written in blue.

> The technical note builds on previous work from Gauch et al, 2021 by an improved multi-time-step / multi frequency LSTM (MF-LSTM) architecture. The MF-LSTM is capable to handle inputs with different temporal resolutions and input variables within a single LSTM cell and provides streamflow predictions on high resolution time steps in the same performance and computational efficiency as Gauch et al, 2021.
>
> I think the paper is well-written, clearly structured and has the potential to advance multi-frequency LSTM applications. I think it fits well to the scope and I support publication as a Technical Note in HESS.

We thank the referee for the well-structured summary of our paper.

> I understand the improvements to the LSTM simplifies the structure and then potentially code maintenance and flexibility of the LSTM code. However, I have difficulties to identify the major added value of the approach to the hydrological community:
>
> The part "enhancing generality and simplicity of use" (Abstract, l.9) is, as I understand it, the main difference between previous work conducted by Gauch et al. 2021 and this Technical Note. I suggest to elaborate more on this point in the paper as, at its current state, it is not clear to me why this is the case. If a single LSTM cell is able to handle the same data and processes as two cells, isn't that single LSTM cell becoming more complex? What is the tradeoff / advantage here? I provided further comments below that might help to understand where I think more details could help in that regard.

Response: Thank you for the suggestion on elaborating why the MF-LSTM enhances generality and simplicity of use.

Concerning the difference with the sMTS-LSTM, our architecture can handle a different number of inputs for each temporal frequency, which gives a clear advantage in an operational setting.

The primary distinction from MTS-LSTM lies in our simpler architecture, which we believe enhances the current state-of-the-art. We demonstrate that a single LSTM cell can effectively manage processes operating at different time scales, eliminating the need for separate LSTM cells for each time scale, plus transfer functions between their hidden states. This results in

a more parsimonious model that aligns more closely in structure and usage with traditional single-frequency LSTM models, making the transition from single-frequency to multi-frequency LSTMs more intuitive for users.

About the idea of the single LSTM cell becoming more complex, there are a couple of things we need to consider. First, defining complexity in this type of model is not straightforward, and current research is being done to compare different architectures using information theory principles. Second, an alternative hypothesis is that there are high similarities between the processes at different timescales (e.g. daily and hourly), and the LSTM architecture can use these similarities plus its ability to regulate the gates based on the current context, to handle the multiple frequencies. Therefore, by handling everything in a single LSTM cell we are being able to leverage information encoded in our data.

Lastly, that the LSTM is being able to handle past forcings at different resolution than more recent ones, without losing performance, underpins that the LSTM cell is acting similar to a process-based hydrological model, with a dissipative behavior when it comes to memory of past forcings. This is a small step towards understanding LSTM based predictions better, given that they are gaining popularity in the hydrology field.

We will include an in-depth discussion of these points in a revised version of the manuscript.

> l.15 - I would add that this is particularly the case for small, fast responding catchments.

Response: Thank you for the suggestion, we will incorporate this in a revised version of the manuscript.

> l.18 - I know that different processes can be at play, but you might want to mention that shorter and flexible time steps are also a prerequisite for eventually being able to depict flash floods, which would also be a strong motivation

Response: Thank you for the suggestion, we will incorporate this in a revised version of the manuscript.

> l.50 - why two weeks? (also l.99). I see, it is mentioned in l.135 - suggest to give that explanation earlier.

Response: The value of 336 hours (14 days) was established by Gauch et al. (2021) through hyper-parameter optimization. We will incorporate this explanation in a revised version of the manuscript.

> l.98 - I acknowledge that the LSTM normalizes the data internally anyway and per se does not 'care' about the pysical plausibility of the inputs vs outputs. But given the

hydrological focus of the journal it might make sense mentioning that it is not needed to use the sum instead of average as an input for precipitation?

Response: Thank you for the suggestion. As you mentioned, using the sum of the average is similar, as the data is normalized internally. We will mention this in a revised version of the manuscript.

l.112-118 This section is important to understand the difference between the MTS-LSTM and the MF-LSTM. However, I find it hard to grasp the structural difference between these two approaches. Can you give more details on the structure of the two different LSTM cells (MTS-LSTM) vs the embedding networks (MF-LSTM)? I suggest to particularly focus on the advantage a user gains. Computationally-wise both approaches are similar as you state later, both approaches can handle the same temporal and variable flexibility and both approaches yield the same performance. For an end-user of your provided codes the question arises why to choose the MF over the MTS-LSTM (with which a user might be familiar already)? Is input data generation and provision simpler - and if yes how?

This paragraph refers to the ability of the MF-LSTM and MTS-LSTM to handle a variable number of inputs for each frequency (which the sMTS-LSTM do not have). In general, an LSTM cell has a predefined number of input channels that are used. In the case of the MTS-LSTM, because different LSTM cells are being used for each temporal frequency, one can input different number of inputs in each cell (e.g. 10 inputs for daily frequency and 20 inputs for hourly frequency).

We propose the use of embedding networks as an alternative solution. In this case, one should define a predefined number of channels that the LSTM cell will use. Then, one uses an embedding network for each frequency (in our case a fully connected layer) to map the number of inputs of each frequency to the predefined number of channels of the LSTM. This strategy allows us to separate the steps of our pipeline. We use the LSTM only for sequence processing, and we use the embedding layers to prepare the original information into the format/type that the LSTM requires.

Concerning the advantages for the user, this would depend on the level of abstraction. On one hand, if the user is only concerned with running the model and recovering the results, without looking at or modifying the code, both approaches would be similar. The only high-level difference would be the understanding that the MF-LSTM is a more parsimonious model. On the other hand, if the user is more research-oriented and is interested in understanding/modifying the code, the MF-LSTM does present a considerable advantage. The code is simpler to understand and, to the MF-LSTM's benefit, the internal functioning is extremely similar to the standard LSTM model. Only some small changes are required in the function that handles the data to the LSTM model (to carry out the averages for the lower frequencies) and also some extra lines in the forward pass of the model in case one uses

embeddings. For this type of users, this greatly facilitates the shift from classical single-frequency LSTM to multi-frequency models.

In a revised version of the manuscript we will include this discussion. We will also include an appendix with an sketch of the MTS-LSTM, so readers can further visualize the differences without having to look for Gauch et al. (2021).

> l.131ff - I think you should provide a few more details about the application of the MF-LSTM: (1) did you conduct a hyperparameter tuning and if yes, how are the hyperparameters comparable to Gauch et al. 2021 (could you provide a small table comparing the hyperparameters)? If you transferred the hyperparameters from the previous study, is that plausible given the different architecture? (2) you don't mention for what time period your results are presented. I assume you show the testing results?

Response: The hyperparameters of the model were adopted from Gauch et al. (2021), who conducted hyperparameter tuning. We acknowledge the referee's concern that transferring these parameters across different architectures may not guarantee optimal model performance. However, the primary objective of this technical note is to introduce the new architecture. Furthermore, we demonstrate that, even with the given hyperparameters, the proposed model achieves performance comparable to the current state-of-the-art. Considering the significant computational cost of performing hyperparameter tuning on these models, we propose to retain the existing setup. In a revised version of the manuscript, we will explicitly acknowledge this as a limitation of the study.

> l.137 - median: would that be the "median streamflow across the 10 models for each time step"? If yes, I would suggest to add this in brackets

Response: Yes exactly. We will indicate this in a revised version of the manuscript.

> l.170 Similar as comment to l.112-118 - you mention that there is no significant difference between processing a batch. I wonder why this is the case. You now have only a single LSTM cell while for the MTS and sMTS you have two. What is then the advantage of your approach over the previous architectures?

Response: In our case, even with a simpler architecture (MF-LSTM), all methods exhibited a similar computational cost. Despite our efforts to standardize as many factors as possible to ensure a fair comparison, certain differences remained. For instance, the MTS-LSTM and sMTS-LSTM were implemented using the Neural Hydrology package, whereas the MF-LSTM was implemented via the Hy2DL package.

Our primary aim was to demonstrate that batch processing times are comparable across methods and, in all cases, faster than the Naive approach. For further details on the advantages of the MF-LSTM, please refer to our responses to the comments above.

Language suggestions:

l.76 - correct "same experimental allowed"

Response: Thank you for the suggestion, we will correct this in a revised version of the manuscript.

l.87 - "observation that" could be replaced by something like "principle" ?

Response: Thank you for the suggestion, we will correct this in a revised version of the manuscript.

l.88 - "reservoir" I find "storage" more appropriate in this context

Response: Thank you for the suggestion, we will correct this in a revised version of the manuscript.

l.91 - "time-varying" wouldn't "time-independent" gating be more appropriate?

Response: Thank you for the suggestion. The gating function is not time-independent, it depends on previous and current time steps. In a revised version of the manuscript, we can change time-varying for time-dependent.

Figure 1 caption : "where one has" suggest to rephrase to "where the same ... are available" or "... exist"?

Response: Thank you for the suggestion, we will correct this in a revised version of the manuscript.

l.167 - "comparing the total training time ... influenced by external factors"? I think you want to say that "total training time is influenced by external factors"?

Response: Thank you for the suggestion, and yes that is what we want to say. We will correct this in a revised version of the manuscript.

Final remarks

We would like to thank the referee for the overall positive evaluation of our manuscript and hope we could address the questions raised in a satisfactory manner.

---

## Author Response (AR1)

**Author´s response**

Dear. Dr. Fenicia

Please find attached with this document the new version of our manuscript, which incorporates the changes suggested during the review process. Attached to this document, you will find the revised manuscript along with a tracked changes version. Below, we provide a summary of the changes made.

**Changes proposed by Anonymous Referee #1.**

Following the feedback of Anonymous Referee 1, and based on our responses from the discussion phase:

- Line 25-26: We modified the sentence.
- Lines 95 -107: We expanded the example's explanation and explicitly indicated that the number of timesteps processed in each frequency is a hyperparameter.
- Line 159: We added a figure in the appendix comparing the full NSE distributions.

**Changes proposed by Anonymous Referee #2.**

Following the feedback of Anonymous Referee 2, and based on our responses from the discussion phase:

- Line 15: We added the requested information.
- Line 18: We added the requested information.
- Line 50: We added the requested information
- Line 98: We added the requested information.
- Line 112-118: We expanded the explanation to further clarify the structural differences between the MTS-LSTM and the MF_LSTM. Moreover, we added a figure in the appendix to illustrate how the MTS-LSTM model works.
- Line 131: In conclusions, we acknowledged as a limitation the fact that we transferred the hyperparameters from Gauch et al. (2021).
- Line 137: We modified the sentence.

Moreover, to further clarify the added advantages that the MF-LSTM provides, with respect to the existing methods, we expanded *Section 2.2* and *Conclusions*

Language suggestions

- Line 76: We modified this sentence.
- Line 87: We modified this sentence.
- Line 88: We modified this sentence.
- Line 91: We modified this sentence.
- Caption Figure 1: We modified the caption.
- Line 167: We modified this sentence.

We believe the modifications made cover the changes proposed by the referees. We would like to thank both referees, as their input in the review process allowed us to produce a better manuscript.

Kind regards,
Eduardo Acuña on behalf of the co-authors.